# Imaging plant germline differentiation within Arabidopsis flowers by light sheet microscopy

Sona Valuchova[1†], Pavlina Mikulkova[1†], Jana Pecinkova[1], Jana Klimova[2], Michal Krumnikl[2,3], Petr Bainar[2], Stefan Heckmann[4], Pavel Tomancak[5], Karel Riha[1]*

[1]Central European Institute of Technology (CEITEC), Masaryk University, Brno, Czech Republic; [2]IT4Innovations, VSB–Technical University of Ostrava, Ostrava, Czech Republic; [3]Department of Computer Science, FEECS VSB – Technical University of Ostrava, Ostrava, Czech Republic; [4]Leibniz Institute of Plant Genetics and Crop Plant Research (IPK), Seeland, Germany; [5]Max Planck Institute of Molecular Cell Biology and Genetics, Dresden, Germany

**Abstract** In higher plants, germline differentiation occurs during a relatively short period within developing flowers. Understanding of the mechanisms that govern germline differentiation lags behind other plant developmental processes. This is largely because the germline is restricted to relatively few cells buried deep within floral tissues, which makes them difficult to study. To overcome this limitation, we have developed a methodology for live imaging of the germ cell lineage within floral organs of Arabidopsis using light sheet fluorescence microscopy. We have established reporter lines, cultivation conditions, and imaging protocols for high-resolution microscopy of developing flowers continuously for up to several days. We used multiview imagining to reconstruct a three-dimensional model of a flower at subcellular resolution. We demonstrate the power of this approach by capturing male and female meiosis, asymmetric pollen division, movement of meiotic chromosomes, and unusual restitution mitosis in tapetum cells. This method will enable new avenues of research into plant sexual reproduction.

*For correspondence:
karel.riha@ceitec.muni.cz

†These authors contributed equally to this work

**Competing interests:** The authors declare that no competing interests exist.

## Introduction

Sexual reproduction in eukaryotes is characterized by the alternation of haploid and diploid life forms. Transitions between these phases are marked by meiosis, the specialized cell division that produces haploid cells from diploid precursors, and fertilization, when haploid gametes fuse and reconstitute the diploid zygote. In multicellular organisms, this process involves differentiated germ cells that form in dedicated organs. In higher animals, the germline segregates from the soma early during embryogenesis, presumably to permit the emergence of complex developmental processes to form a staggering variety of cells and organs, while protecting the cells destined for reproduction from mutation load (*Kumano, 2015*; *Radzvilavicius et al., 2016*). In contrast to animals, the germline in higher plants differentiates from somatic cells at late stages of the life cycle. In angiosperms, gametogenesis occurs in the anthers and ovaries of developing flowers, resulting in the formation of pollen and the embryo sac that harbor male and female gametes, respectively.

Plant germline differentiation includes a number of remarkable cellular events. The formation of the male germ cell lineage begins with the differentiation of pollen mother cells (PMCs) from mitotically amplified sporogenous cells. PMCs undergo meiosis, a reductional cell division in which paired homologous chromosomes segregate in meiosis I and sister chromatids in meiosis II. Chromosome segregation during meiosis requires extensive remodeling of the cell cycle machinery to permit

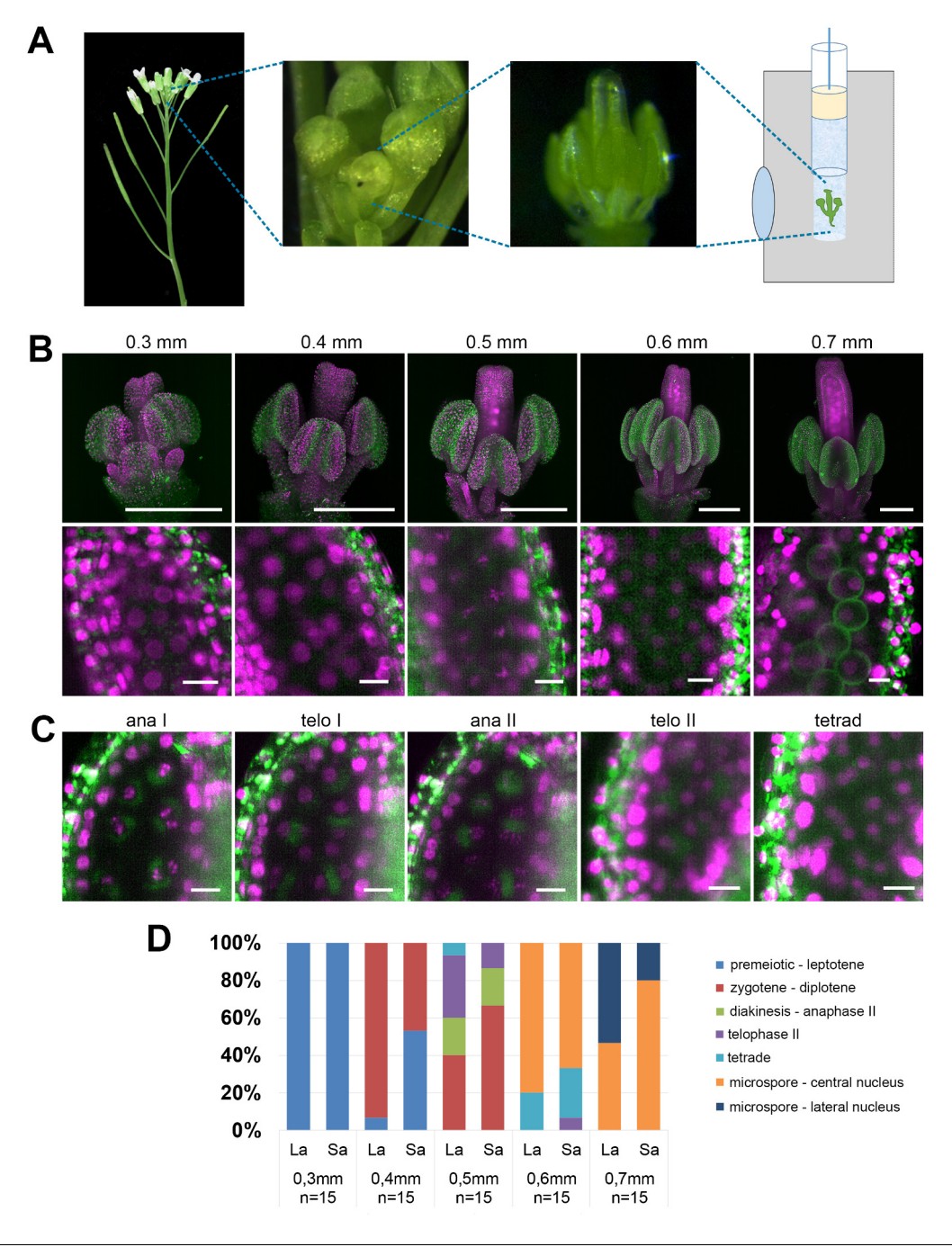

**Figure 1.** Imaging Arabidopsis flower using LSFM. (**A**) Workflow of sample preparation. (**B**) Maximum intensity projections (MIPs) of micrographs of HTA10:RFP flowers dissected from buds of the indicated sizes (upper panel, scale bar 200 µm). A detailed image of a single anther lobe with PMCs and microspores is shown in the lower panel (scale bar 10 µm). HTA10:RFP in magenta, 488 nm autofluorescence in green. (**C**) Examples of additional meiotic stages. Scale bar 10 µm. (**D**) Distribution of meiotic stages from premeiosis/leptotene to microspores with laterally located nuclei in floral buds of different sizes. One long anther (La) and one short anther (Sa) were analyzed from each floral bud. The frequency of different meiotic stages was estimated from 15 buds of the same width.

The online version of this article includes the following figure supplement(s) for figure 1:

**Figure supplement 1.** Growth dynamics of a floral bud.

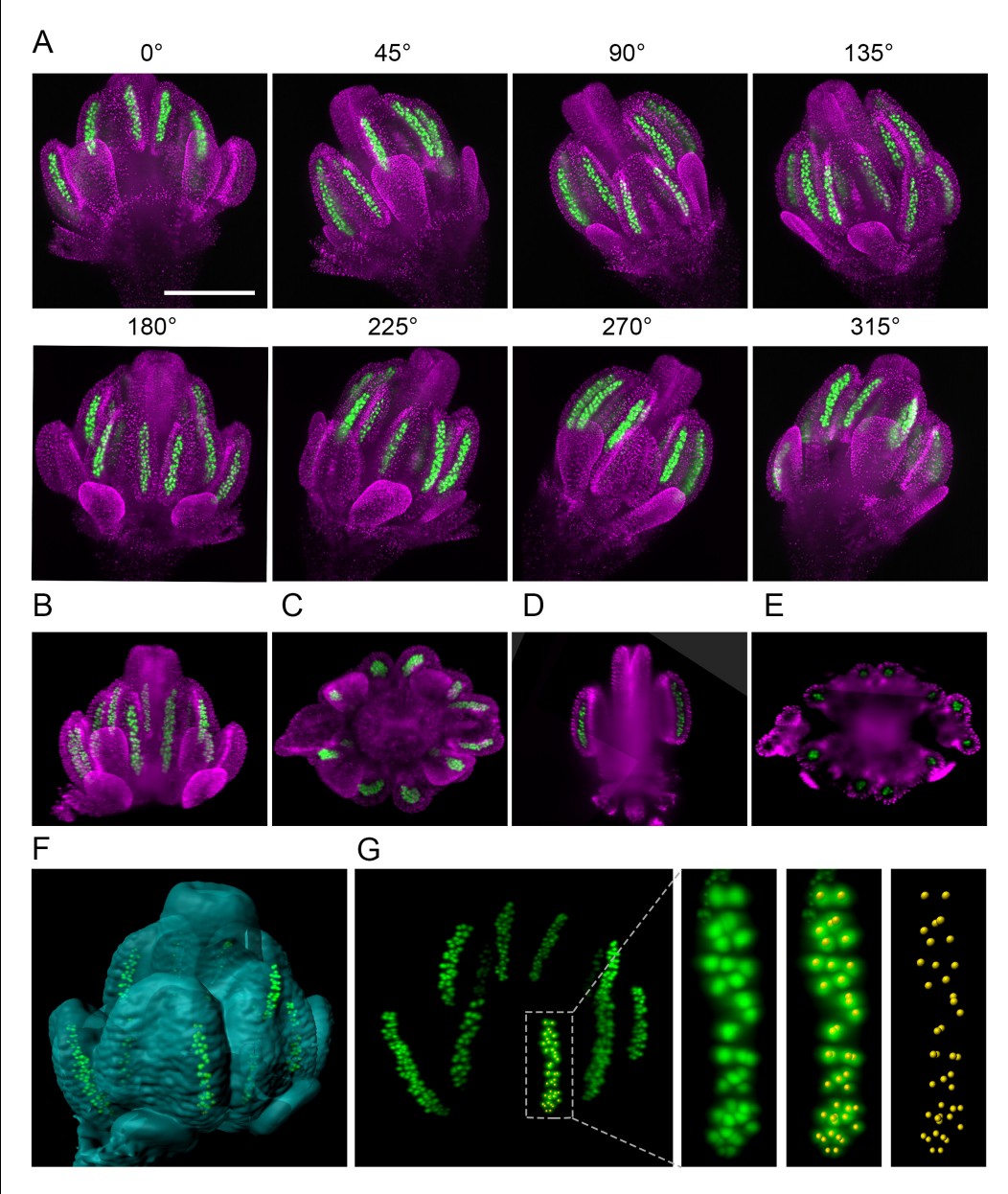

**Figure 2.** The 3D reconstruction of Arabidopsis flower from multiview imaging. (**A**) MIPs of a 0.5 mm floral bud expressing ASY1:eYFP (green) and H2B:mRuby2 (magenta) viewed from eight different angles. Scale bar 200 µm. (**B,C**) Imaris MIP of 3D reconstructed flower. Longitudinal (**D**) and transversal (**E**) sections of the 3D reconstructed flower. (**F**) Surface rendered 3D model of the flower with indicated PMCs. (**G**) MIP of PMCs from the 3D model. Automated detection of PMCs using Imaris spot detection in one anther lobe is shown (41 PMCs were counted).

The online version of this article includes the following video for figure 2:
**Figure 2—video 1.** Animation of 3D reconstructed flower expressing ASY1:eYFP (green) and H2B:mRuby2 (magenta).
https://elifesciences.org/articles/52546#fig2video1

recombination of homologous chromosomes during the extended prophase I, co-segregation of sister chromatids and protection of centromeric cohesion in anaphase I, and inhibition of DNA replication in interkinesis (*Marston and Amon, 2004*; *Petronczki et al., 2003*). After meiosis, haploid microspores divide through asymmetric mitosis producing a highly compact generative nucleus and a diffuse vegetative nucleus. This is thought to coincide with epigenetic reprograming of the

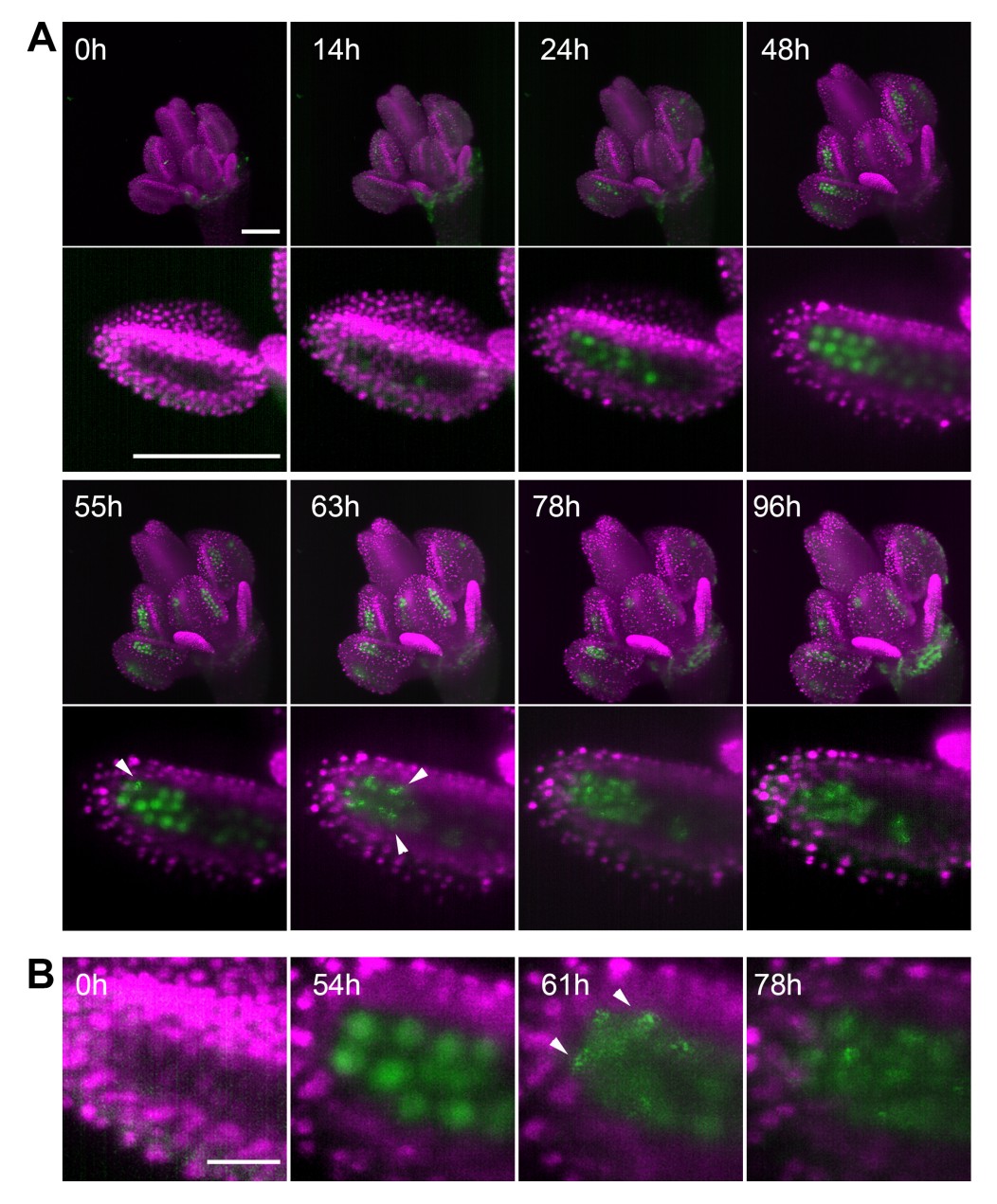

**Figure 3.** Time lapse imaging of a growing flower. (**A**) MIPs of a flower (upper panel) expressing ASY1:eYFP (green) and H2B:mRuby2 (magenta) and one of its anther lobes (bottom panel) at indicated time points (scale bar 100 μm). (**B**) Detailed view of the distribution of ASY1 signal in developing PMCs. Scale bar 20 μm. Arrowheads indicate ASY1 speckles.

The online version of this article includes the following video and figure supplement(s) for figure 3:

**Figure supplement 1.** Development of a floral bud in the closed capillary.

**Figure 3—video 1.** Time lapse imaging of floral bud development in 60 min intervals (ASY1:eYFP in green, H2B: mRuby2 in magenta).

https://elifesciences.org/articles/52546#fig3video1

generative nucleus, which subsequently divides to produce two identical sperm cells (*Schmidt et al., 2015*). In the female germline, one archesporial cell per ovule differentiates into a so called megaspore mother cell (MMC) that enters meiosis. Three of the four resulting haploid spores degenerate, while the remaining one undergoes three mitotic divisions to form an syncytial female gametophyte

with eight nuclei (*Schmidt et al., 2015*). Further cellularization produces seven-celled mature embryo sac containing two female gametes, two synergids and three antipodal cells.

Plant reproduction has been extensively studied since the dawn of modern biology. Genetic approaches combined with traditional morphology, histology, and cytogenetics have provided a wealth of data on the genetic framework governing sexual reproduction in model plant species, with the most comprehensive pictures available in Arabidopsis, maize, and rice (*Mercier et al., 2015*). This information has been further expanded by transcriptomics and proteomics, though use of these approaches is hindered by difficulties in obtaining pure fractions of germ-cells in sufficient quantities (*Honys and Twell, 2003*; *Zhou and Pawlowski, 2014*; *Sánchez-Morán et al., 2005*). Recent advances in single cell transcriptomics promise break-throughs also on this front (*Nelms and Walbot, 2019*). Nevertheless, further understanding of germline differentiation requires methods allowing description of the accompanying molecular and cellular processes with high spatial and temporal resolution. Live cell imaging represents an important tool for capturing dynamics of cell growth and division, protein expression and localization, as well as responses to environmental and genetic perturbations (*Roeder et al., 2011*).

Live cell imaging has been instrumental in research into root growth and development. Due to its simple structure with stereotypical cell patterning, transparency, and ease of cultivation within microscopy chambers, the root has become a key model for cell biology in plants (*Ovečka et al., 2018*; *Grossmann et al., 2018*). In contrast, relatively little has been done with time-lapse microscopy of cellular processes within the flower. This is because flower imaging is technically much more challenging. Flowers develop on adult plants and reproductive tissues are hidden deep within the floral organs. This challenge has been partially overcome by using semi in vitro systems in which reproductive tissues such as ovules and anthers were dissected from flowers, grown on cultivation media and examined by confocal or multi-photon microscopy (*Kurihara et al., 2013*). This approach enabled live imaging of double fertilization, early embryogenesis, and chromosome pairing during male meiosis (*Hamamura et al., 2011*; *Gooh et al., 2015*; *Feijó and Cox, 2001*; *Sheehan and Pawlowski, 2009*). However, these semi in vitro systems have several limitations including a low survival rate of excised tissues, aberrant development which limits duration of live imaging, and altered physiology due to in vitro cultivation that may affect some cellular processes (*Kurihara et al., 2013*). Thus, there is a need to develop imaging methods in the context of whole organs or plants. Confocal live imaging of emerging floral buds attached to a short stem embedded in media was described for Arabidopsis (*Prunet et al., 2016*). Recently, this protocol was modified for live cell imaging of male meiosis in which two anthers in larger buds were exposed to microscopy by removal of a sepal (*Prusicki et al., 2019*). This allowed observation of PMCs for up to 48 hr, although movies longer than 30 hr were usually not informative due to loss of the focal plane (*Prusicki et al., 2019*).

Light sheet fluorescence microscopy (LSFM) has emerged as a powerful imaging technique for real time visualization of complex developmental processes at subcellular resolution (*Keller, 2013*). In LSFM, a sample is excited with a thin sheet of laser light and the generated single optical section is captured by a perpendicularly oriented detection lens. This arrangement results in low phototoxicity (*Icha et al., 2016*), because only the plane of the specimen that is imaged gets illuminated. 3-dimensional data (3D) are obtained by moving the sample through the light sheet. In some types of LSFM, the sample can be rotated in front of the detection lens to acquire 3D image data of the same specimen from multiple angles (*Huisken et al., 2004*). The LSFM acquisitions are invariably fast because of the entire illuminated plane is captured at once with digital cameras. In combination, these features allow long term imaging of highly dynamic cellular processes within complex biological samples such as developing embryos and organs (*Ovečka et al., 2018*; *Weber and Huisken, 2011*). In plants, LSFM has mainly been used for imaging root growth and development with only sporadic attempts in other plant structures, such as seedlings and flowers (*Maizel et al., 2011*; *Ovečka et al., 2015*; *Ovečka et al., 2018*;

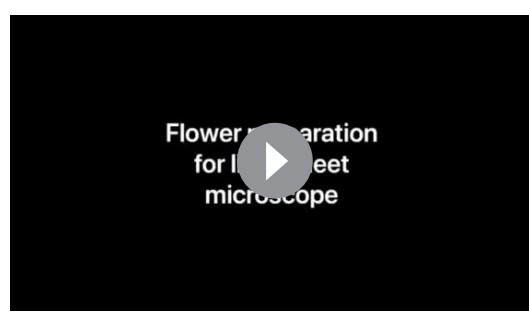

**Video 1.** Preparation of a sample for imaging by LSFM.
https://elifesciences.org/articles/52546#video1

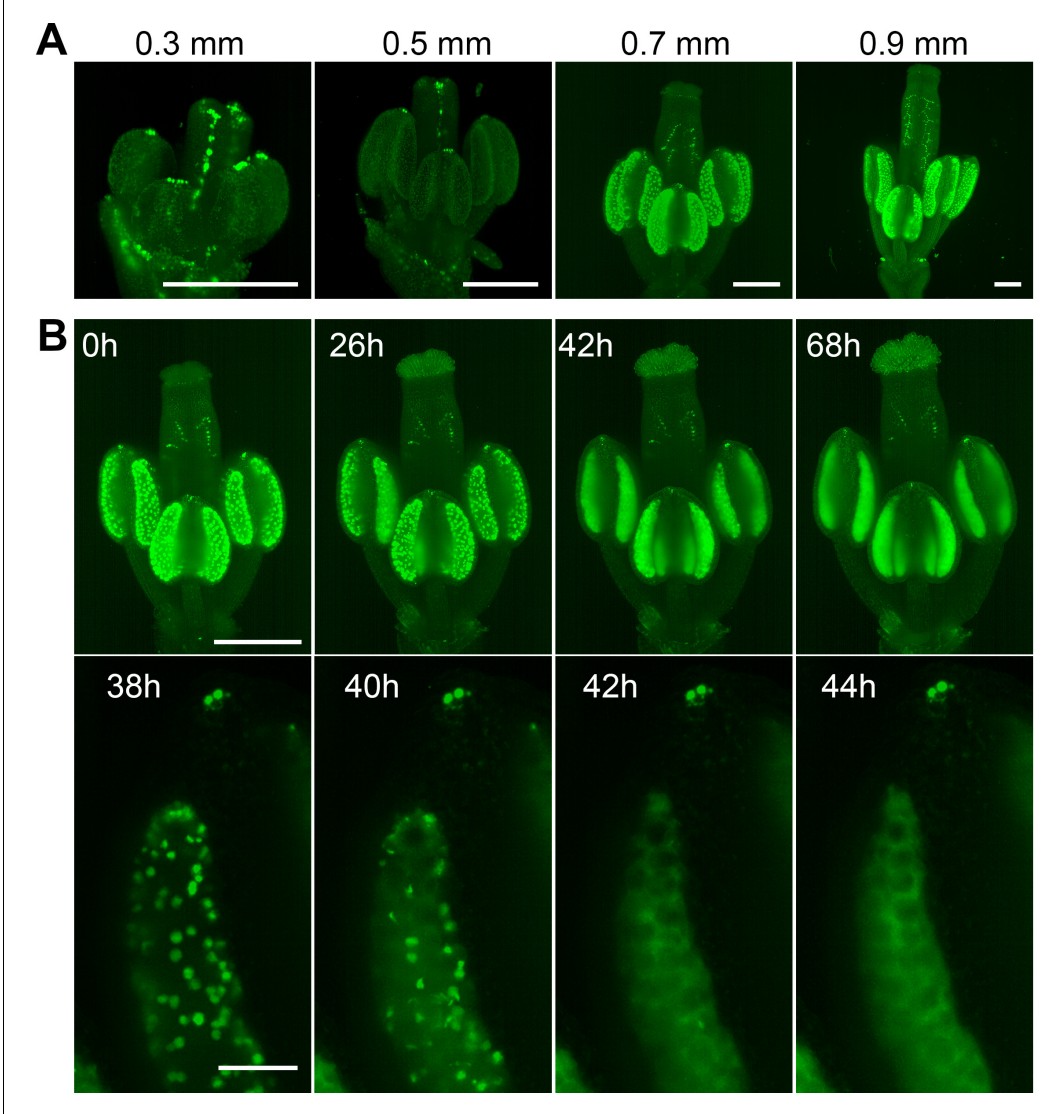

**Figure 4.** Spatiotemporal distribution of auxin response in flower. (**A**) MIPs of *DR5::N7-Venus* signal of four different flower buds of different sizes. Scale bar 200 µm. (**B**) Time lapse imaging of a flower at developmental stage 12 expressing *DR5::N7-Venus* in 2 hr intervals (upper panel). Scale bar 300 µm. Lower panel: detail of a loculus showing release of the nuclear content into the cytoplasm between 40 and 42 hr. Scale bar 50 µm. Non-linear transformation by gamma was used to enhance the outline of the flower.

The online version of this article includes the following video for figure 4:

**Figure 4—video 1.** Time lapse imaging of a flower at developmental stage 12 expressing *DR5::N7-Venus* in 2 hr intervals.

https://elifesciences.org/articles/52546#fig4video1

*Grossmann et al., 2018*). Here we describe the establishment of live cell imaging of Arabidopsis flowers and its applications in investigating diverse aspects of plant germline differentiation.

## Results

### 3D-reconstitution of the Arabidopsis flower at cellular resolution

To assess the suitability of LSFM for imaging cellular processes within flowers, we decided to establish a protocol for imaging male meiosis using a lightsheet microscope. Meiosis is cytologically the

most easily distinguishable stage of plant germline differentiation. An Arabidopsis flower contains approximately 600 PMCs within six anthers, each harboring four loculi. Meiosis is highly synchronous within a loculus and its onset is tightly coupled to development, typically occurring at flower developmental stage 9 (*Sanders et al., 1999*; *Ma, 2006*). At this stage, reproductive organs are fully enclosed by sepals while petals are still relatively rudimentary structures that do not cover the anthers (*Smyth et al., 1990*). To visualize meiosis, we used an Arabidopsis line harboring an *HTA10: RFP* reporter construct that marks chromatin with fluorescently tagged histone H2A (*Dumur, 2019*) (Yelagandula and Berger, personal communication, 2019). Growth and development of flowers in plants carrying the construct is comparable to wild type (*Figure 1—figure supplement 1*).

Floral buds ranging in width from 0.3 to 0.7 mm were detached from inflorescences at the main inflorescence bolt, sepals were carefully removed to expose anthers, and flowers were embedded in low melting point agarose within a capillary (*Figure 1A*) (*Ovečka et al., 2015*). The capillary was attached to a holder in the microscope and the capillary was rotated to find the best angle for imaging. LSFM showed that the *HTA10:RFP* reporter is uniformly expressed in nuclei throughout the entire flower, including PMCs (*Figure 1B*). Although we used only 10x objectives for imaging, this magnification was sufficient to distinguish the major meiotic stages (*Figure 1B,C*). Determination of the cell cycle stage was further guided by autofluorescence in the GFP channel, which allowed visualization of certain cellular landmarks such as the organellar band and cell wall in microspores. We next determined how the width of a floral bud correlates with the presence of individual meiotic stages. This is a very practical parameter which enables pre-selection of appropriate buds based on their size prior to continuing with the rather laborious sample preparation. LSFM micrographs showed that 0.3 mm buds contained mainly pre-meiotic or leptotene PMCs, meiotic divisions were detected in 0.5 mm buds, and microspores with centrally or laterally localized nuclei were present in 0.7 mm buds (*Figure 1D*). Two of the six anthers in Arabidopsis are shorter than the others. Our staging experiment indicated that meiosis in the shorter anthers is slightly delayed compared to the remaining four.

One of the key applications of LSFM is 3D reconstruction of larger biological specimens from multiple views. To reveal the structure of entire Arabidopsis flowers at subcellular resolution, we generated an Arabidopsis line harboring *H2B:mRuby2* for visualization of somatic nuclei and *ASY1:eYFP* that is expressed specifically in meiocytes. We scanned the flower from eight views differing by 45° increments (*Figure 2A*). Signals from nuclei were used to register all views by the Fiji Multiview reconstruction plugin (*Preibisch et al., 2008*). The registered data were combined into a single output image using a weighted average fusion implemented in the Multiview reconstruction plugin (*Figure 2B–F*, *Figure 2—video 1*). The resulting 3D-model of the flower showed that while outer structures, such as developing petals and the outer loculi of anthers, were clearly visible, LSFM did not provide sufficient penetration to resolve inner loculi and the pistil. Nevertheless, the 3D model can be rotated to display features that are not apparent from individual scans. For example, we used the ASY1 signal to determine the 3D arrangement of PMCs in outer loculi and applied the spot detection wizard in Imaris to count them automatically (*Figure 2G*). The 3D reconstruction permits precise quantification of male germ cells in anther lobes, and in combination with time-lapse imaging, it will enable tracking differentiation of individual germ cells.

## Live imaging of flower development

Experiments described in the previous section demonstrated that LSFM provides sufficient depth and resolution to capture subcellular events in the male germline. Our next goal was to establish live cell imaging for examining different stages of germline differentiation in the developing flower. The male germline separates from other cell lineages with the formation of sporogenous cells at flower developmental stage seven and its differentiation is completed with the second pollen mitosis resulting in trinuclear pollen at stage 12 (*Sanders et al., 1999*). Thus, the entire development of the male germline lasts approximately 7 days, based on the duration of individual stages as determined by *Smyth et al. (1990)*.

To continuously image germ cells over several days, we had to overcome several technical obstacles. First, we empirically determined that proper development of the detached flower within the capillary requires media with a high sugar content. However, the presence of such rich media in the microscopy chamber quickly led to contamination. We solved this problem by cultivating flowers in a sealed capillary that was submerged in 6% glycerol in the microscopy chamber. The glycerol

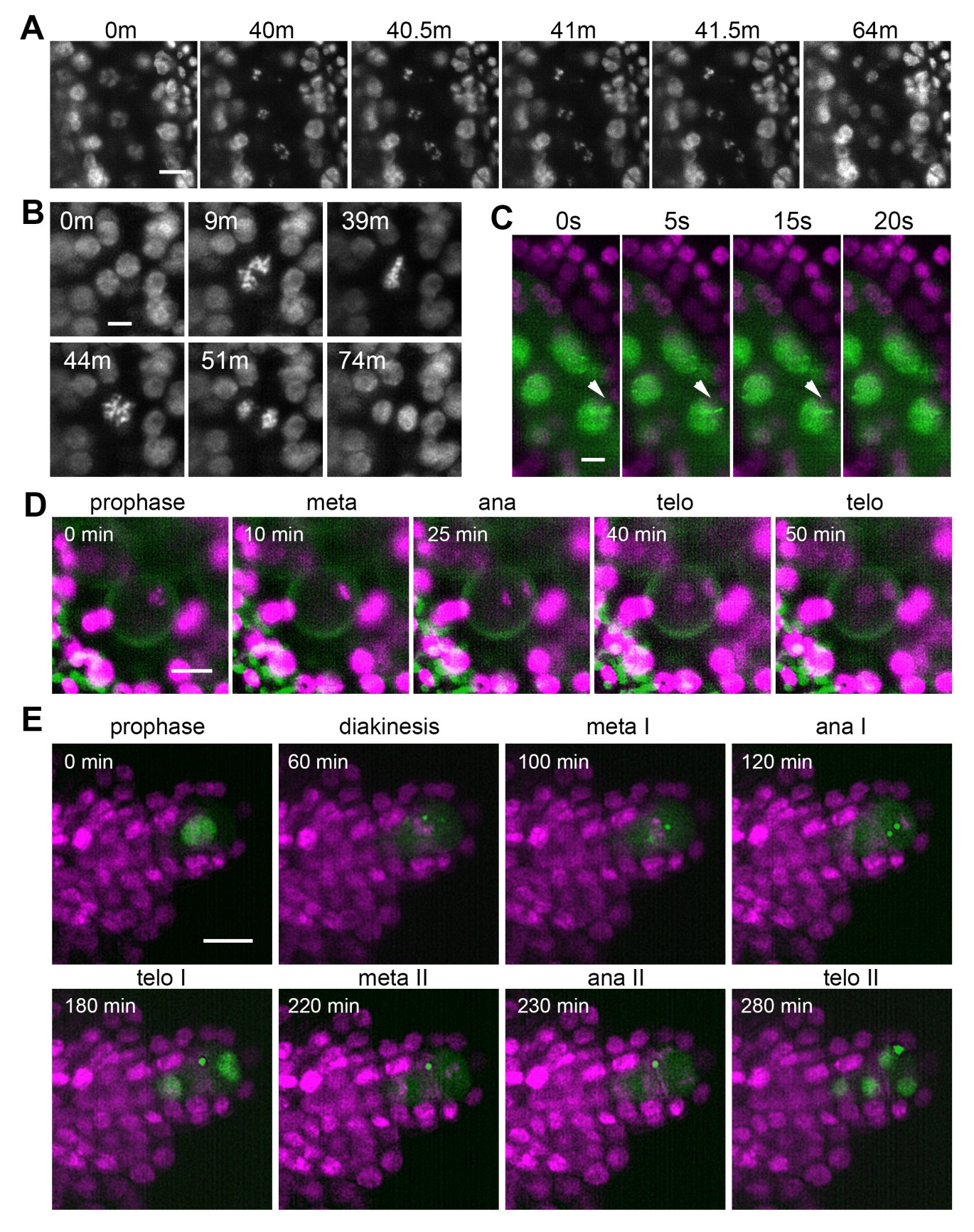

**Figure 5.** Time lapse imaging of subcellular processes within the flower. (**A**) Chromosome segregation in meiosis I from diakinesis (0 m) to telophase I (64 m) visualized with the *HTA10:RFP* marker. Images were taken every 30 s, scale bar 10 μm. (**B**) Restitution mitosis in tapetum cells. Images were taken every 60 s, scale bar 5 μm. (**C**) Rapid chromosome movements in zygotene. Chromatin axes are visualized with ASY1:eYFP (green), somatic nuclei with H2B:mRuby2 (magenta). Arrowhead points to a chromatin axis that moves within the indicated interval. Images were taken every 5 s, scale bar 5 μm. (**D**) Asymmetric pollen mitosis I. Chromatin is visualized with HTA10:RFP (magenta), 488 nm autofluorescence highlights the pollen wall (green). Images were taken every 5 min, scale bar 10 μm. (**E**) Female meiosis. MMC is marked with ASY1:eYFP (green), chromatin with HTA10:RFP (magenta). Images were taken every 10 min, scale bar 10 μm.

The online version of this article includes the following video(s) for figure 5:

**Figure 5—video 1.** Time lapse imaging of chromosome segregation in PMCs from diakinesis through telophase II in 30 s intervals.
https://elifesciences.org/articles/52546#fig5video1

**Figure 5—video 2.** Time lapse imaging of restitution mitosis in tapetum cells in 60 s intervals.
https://elifesciences.org/articles/52546#fig5video2

**Figure 5—video 3.** Time lapse imaging of asymmetric pollen mitosis I in 5 min intervals.
https://elifesciences.org/articles/52546#fig5video3

**Figure 5—video 4.** Rapid movements of chromatin axes in zygotene in 5 s intervals.
https://elifesciences.org/articles/52546#fig5video4

**Figure 5—video 5.** Time lapse imaging of female meiosis in 10 min intervals.
https://elifesciences.org/articles/52546#fig5video5

served to equalize the refractive index of the media in the chamber with the cultivation media in the capillary; using only water resulted in suboptimal images. Under these conditions, we were able to image Arabidopsis floral buds for up to five days (*Figure 3*, *Figure 3—video 1*). To capture the entire male meiosis, we started with 0.3 mm-wide floral buds of the *H2B:mRuby2 ASY1:eYFP* reporter line (*Figure 3A*, 0 h) and performed continuous imaging in 1 hr increments for over 4 days. ASY1 is expressed in early meiosis where it associates with the axial elements of prophase I chromatin. By diplotene it is depleted from chromatin, forming cytoplasmic aggregates (*Armstrong et al., 2002*). In our LSFM experiment, the ASY1 signal appeared after approximately 24 hr of imaging and prominently stained PMC nuclei for the next 30 hr. At 55 hr, ASY1 began to form cytoplasmic speckles that persisted in the cytoplasm beyond cytokinesis and tetrad formation at 78 hr (*Figure 3A,B*, *Figure 3—video 1*). This data demonstrates that meiosis was initiated and successfully completed under our experimental conditions. We observed a gradual enlargement of floral organs over the entire period of imaging, indicating that the detached flowers were able to grow and develop within the microscopy chamber. We determined that long-term LSFM imaging has a negligible effect on meiotic progression by comparing imaged flowers with non-imaged controls cultivated under the same conditions (*Figure 3—figure supplement 1*). Furthermore, this experiment showed that the growth of floral buds cultivated in the capillary is only slightly delayed relative to flowers that developed on plants (*Figure 3—figure supplement 1*).

To further explore the applicability of LSFM for studying other aspects of plant floral development, such as hormone signaling, we analyzed the spatiotemporal distribution of the nuclear-localized auxin response marker *DR5::N7-Venus* (*Wabnik et al., 2013*). Images of flowers taken at different stages of development showed a prominent signal in vascular tissues within the pistil and a massive activation of auxin signaling in the tapetum in postmeiotic anthers (*Figure 4A*). This is consistent with published data on localized auxin synthesis by tapetum cells prior to pollen maturation (*Cecchetti et al., 2008*; *Yao et al., 2018*). Furthermore, we detected strong auxin signaling in approximately four cells at the very tips of the anthers. The tapetum forms the most inner cell layer within the loculus; it provides nutritive support to PMCs and undergoes developmental programmed cell death (PCD) when pollen mitotic divisions occur (*Sanders et al., 1999*; *Parish and Li, 2010*). Live imaging of post-meiotic *DR5::N7-Venus* flowers for more than 3 days recorded the occurrence of PCD as it spreads within individual loculi (*Figure 4B*, *Figure 4—video 1*). Continuous growth of the pistil throughout the duration of the experiment suggests that the observed release of nuclear content into the cytoplasm is not an artifact of cultivation and indeed represents PCD. These two examples of imaging male meiosis and PCD in tapetum demonstrate the power of LSFM for organ-scale analyses of cellular processes in flowers on the timescale of days.

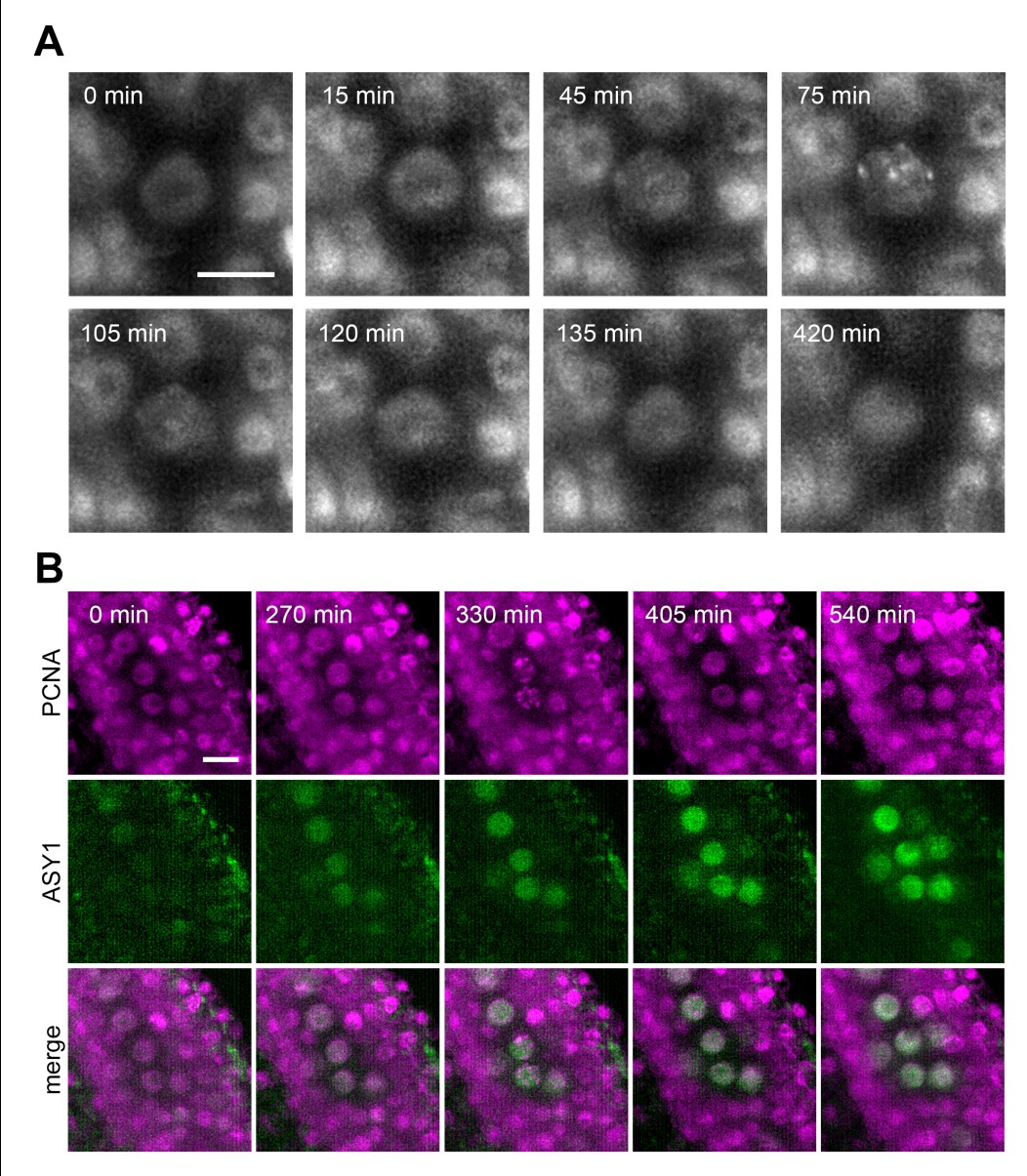

**Figure 6.** Protein localization in meiotic S-phase. (**A**) Time lapse imaging of PCNA:TagRFP during meiotic S-phase. Nuclear speckles are visible between 45 to 120 min. Images were taken every 15 min, scale bar 10 μm. (**B**) Time lapse imaging of PCNA:TagRFP (magenta) and ASY1:eYFP (green) in PMCs. Timeframe ranges from the first appearance of ASY1 signal prior to S-phase (0 min) to late leptotene/zygotene (540 min). Images were taken every 15 min, scale bar 10 μm.

The online version of this article includes the following video(s) for figure 6:

**Figure 6—video 1.** Time lapse imaging of PCNA:TagRFP in PMCs in 15 min intervals.
https://elifesciences.org/articles/52546#fig6video1

**Figure 6—video 2.** Time lapse imaging of PCNA:TagRFP (magenta) and ASY1:eYFP (green) in PMCs in 15 min intervals.
https://elifesciences.org/articles/52546#fig6video2

## Live imaging of subcellular processes within the flower

Next, we asked whether LSFM can provide sufficient spatiotemporal resolution for capturing relatively rapid processes, such as chromosome segregation or movement during meiotic prophase I. Using the *HTA10:RFP* line we were able to visualize the segregation of meiotic chromosomes in the

entire anther lobe with time increments of 30 s (*Figure 5A*, *Figure 5—video 1*). From prometaphase I until telophase II, chromosome segregation lasted for about 130 min. While meiosis is highly synchronous within a loculus, we noticed a temporal gradient in the onset of chromosome segregation across the loculus with PMCs at the tip being approximately 4 min delayed compared to PMCs at the base of the anther (*Figure 5—video 1*).

Tapetum cells in Arabidopsis are binuclear and usually undergo further polyploidization through an unknown mechanism (*Weiss, 2001*). Live imaging revealed that tapetum cells undergo restitution mitosis (*Figure 5B*, *Figure 5—video 2*). This is a unique cellular process in which binuclear cells enter mitosis but metaphase chromosomes from the two nuclei form a single metaphase plate; the subsequent anaphase results in two nuclei with a duplicated set of chromosomes (*Oksala and Therman, 1977*). The restitution mitosis in tapetum cells occurred asynchronously over an approximately 4 hr window that coincides with the diplotene stage of PMCs.

Another remarkable event in germline differentiation is the asymmetric pollen mitosis I that occurs approximately 2–3 days after completion of meiosis (*Sanders et al., 1999*; *Smyth et al., 1990*). Because of its poorly defined timing and relatively short duration, pollen mitosis I is difficult to detect by classical cytology in fixed tissues. Entry into pollen mitosis I is preceded by the movement of microspore nuclei from a central to a lateral position. We managed to capture pollen mitosis I by imaging anthers harboring microspores with laterally located nuclei. It occurs relatively synchronously in neighboring microspores and lasts for about 40 min (*Figure 5D*, *Figure 5—video 3*). Chromosomes that segregate towards the cell periphery form the generative nucleus, while the less condensed vegetative nucleus arises from chromosomes that move to the cell interior.

To further explore the utility of LSFM for imaging rapid processes within a flower, we examined the movements of meiotic chromatin that occur during zygotene and which facilitate the pairing of homologous chromosomes (*Link and Jantsch, 2019*; *Sheehan et al., 2013*). We were able to image movements of axial elements marked with *ASY1:eYFP* by scanning one layer of PMCs in 5 s intervals (*Figure 5C*, *Figure 5—video 4*). This experiment highlights two key advantages of LSFM: the ability to capture large cell volumes within relative short time intervals and limited photobleaching; even after 20 min of imaging we did not notice a substantial loss of signal.

While plant meiosis is an intense area of research, most studies focus on male meiosis while female meiosis remains mostly ignored. There are roughly 50 ovules in the Arabidopsis flower, each of which carries a single MMC that can only be distinguished from the surrounding somatic cells by its central location within the nucellus at the tip of the developing ovule (*Pinto et al., 2019*). The rarity of MMCs together with their morphological resemblance to somatic cells makes female meiosis experimentally less accessible than male meiosis. To overcome this limitation, we aimed to develop a protocol for live imaging of female meiosis. Because developing ovules are inside ovaries under multiple layers of cells, they are invisible to LSFM in the context of an entire flower (*Figure 2*, *Figure 2—video 1*). Therefore, we used exposed ovules for imaging. MMCs differentiate later than PMCs and female meiosis occurs at floral stage 11 (*Schneitz et al., 1995*). We dissected 0.85 mm floral buds by carefully detaching all sepals, petals, and stamens, cut off the stigma, and removed valves to expose ovules attached to the septum. Dissected flowers were embedded in low melting point agarose within a capillary and imaged for up to 24 hr in 10 min increments. We could readily detect MMCs due to the presence of the *ASY1:eYFP* signal and were able to record the first and second meiotic divisions (*Figure 5E*, *Figure 5—video 5*). Under our imaging conditions, meiotic divisions lasted approximately 3.5 hr from metaphase I to telophase II. The ASY1 signal remained detectable for up to 8 hr after the formation of haploid nuclei.

## Use of live imaging in gene function studies

There are two major applications for live cell imaging in gene function studies: spatiotemporal protein localization and detailed analysis of mutant phenotypes. To illustrate the utility of LSFM in protein localization studies, we analyzed the timing of ASY1 expression relative to S-phase. S-phase in Arabidopsis can be monitored by proliferating cell nuclear antigen (PCNA). PCNA exhibits a disperse nuclear localization throughout the cell cycle but forms nuclear foci ranging from small dots to large nuclear speckles during S-phase (*Yokoyama et al., 2016*). Live imaging of pre-meiotic cells expressing *ASY1:eYFP* and *PCNA:TagRFP* markers showed a reorganization of PCNA from a diffused signal to nuclear speckles, which were detectable for approximately 90 min (*Figure 6A*, *Figure 6—video 1*). Cells with speckles likely represent later S-phase. Cells in early S-phase,

characterized by small PCNA dots, could not be clearly distinguished under our imaging conditions. Nevertheless, the duration of the speckle-stage is comparable to mitotic cells (*Yokoyama et al., 2016*), indicating that the pre-meiotic S-phase is not substantially longer than the S-phase of mitotic cells. This is in contrast to observations in other organisms, where pre-meiotic S-phase was reported to be at least twice as long as mitotic S-phase (*Jaramillo-Lambert et al., 2007*; *Blitzblau et al., 2012*). Furthermore, we noticed the appearance of PMCs with laterally localized nucleoli, which is a feature typical for late leptotene and zygotene (*Prusicki et al., 2019*), approximately 3.5 hr after detecting PCNA speckles, indicating that S-phase is immediately followed by meiotic prophase I. The ASY1 signal was detected approximately 5.5 hr ahead of PCNA speckles (*Figure 6B*, *Figure 6— video 2*), demonstrating that ASY1 is expressed before the onset of pre-meiotic S-phase. This experiment shows that LSFM provides sufficient sensitivity and resolution to monitor protein expression and subcellular localization.

Cytogenetic analysis of fixed samples is the key tool for phenotypic characterization of meiotic mutants in plants. However, understanding meiotic defects may be a formidable task as reconstitution of meiotic progression from fixed samples without the knowledge of temporal context is tedious and in some cases even impossible. Previously, we reported that inactivation of the nonsense mediated RNA decay factor SMG7 leads to an unusual meiotic arrest in anaphase II (*Riehs et al., 2008*; *Capitao et al., 2018*). This conclusion was based on a laborious cell-cycle staging experiment that involved the cytogenetic analysis of thousands of PMCs (*Riehs et al., 2008*). Here, we used live cell imaging to reinvestigate meiosis in Arabidopsis *smg7-1* mutants (*Figure 7*). We confirmed that PMCs in *smg7-1* mutants indeed arrest in an irregular anaphase II and do not form haploid nuclei like wild type (*Figure 7*, *Figure 7—videos 1* and *2*). However, we also detected a loculus where PMCs entered telophase II but after approximately 25 min chromosomes recondensed and formed figures resembling irregular anaphase II (*Figure 7*, *Figure 7—video 3*, referred to as anaphase III). Such behavior of PMCs was not detected in fixed tissues by cytology. This demonstrates the power of live cell imaging in discovering new phenotypes even in extensively characterized mutants.

## Discussion

In this study, we describe the utilization of LSFM for live imaging of cellular processes within Arabidopsis flowers. Our protocol enables visualization of an entire floral bud with subcellular resolution over the period of days. One of the major hurdles in long-term live imaging of multicellular structures is their growth out of the field of view. For example, rapid growth of the root tip, the most favorite cell biology model in plants, limits live imaging by conventional microscopes to several hours. Longer imaging requires more sophisticated solutions with automated tracking (*von Wangenheim et al., 2017*). In this respect, developing Arabidopsis flower represents an attractive system to study differentiation of an entire plant organ at the subcellular level. Development of an Arabidopsis flower, from the emergence of the floral primordia until flower opening, takes less than two weeks. During this period, a majority of the cell- and organ-differentiation processes, including formation of male and female germlines, are completed (*Sanders et al., 1999*; *Smyth et al., 1990*; *Schneitz et al., 1995*). A mature floral bud still fits within the field of view of a standard light sheet microscope, which enables continuous recording of long segments of flower development. We routinely performed continuous imaging for up to five days, which is sufficient to capture the entire male and female sporogenesis or the differentiation of tapetum cells.

A great advantage of LSFM is its flexibility in terms of sample positioning and multiview imaging. This permits the position of the sample to be adjusted in order to acquire the best view of the area of interest. Furthermore, multiview imaging increases the chance of successfully recording the sample even when its orientation changes upon growth during experiment. In addition, it allows 3D models to be built, which can enhance features that are not apparent from a single view. We implemented multiview imaging to reconstruct a 3D model of the Arabidopsis flower at unprecedented resolution and used it to extract information on the number of PMCs within a loculus. When combined with time lapse recording, this approach results in sufficiently detailed datasets for quantitative analysis of 4D morphology of flower differentiation at the cellular level (*Bassel and Smith, 2016*). We found that the 10x objective provides sufficient resolution to visualize Arabidopsis subcellular structures, such as chromosomes, PCNA speckles and axial elements of paired chromosomes. If needed, the resolution can further be increased by using higher magnification objectives (20x).

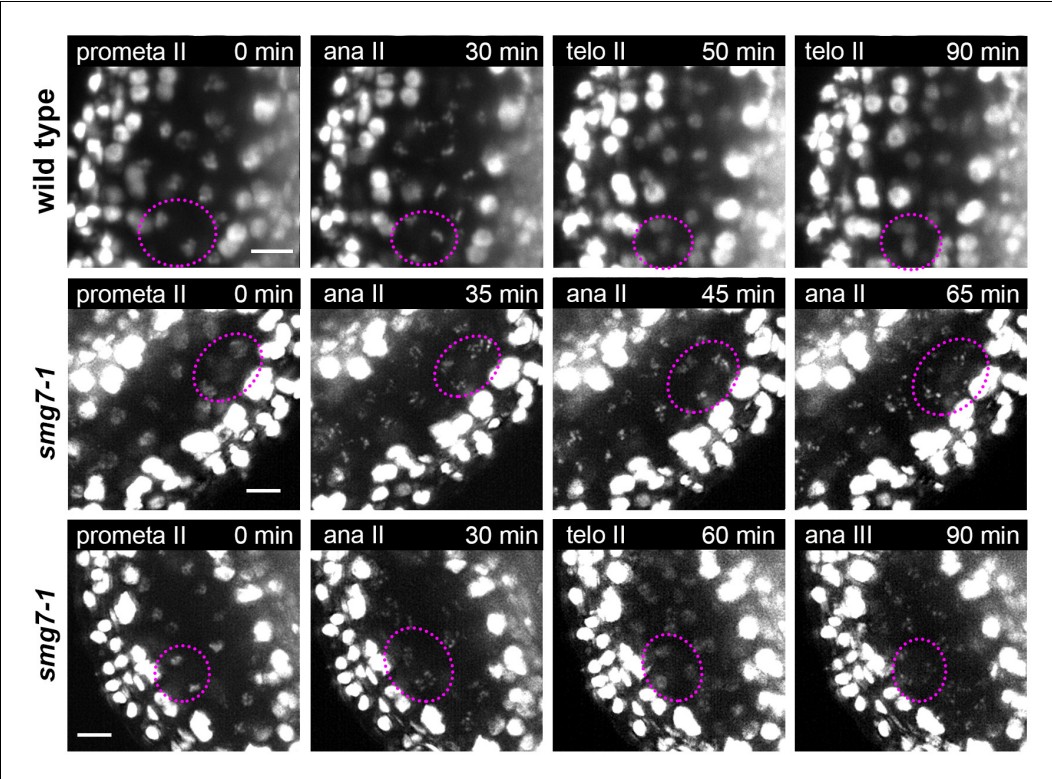

**Figure 7.** Time lapse imaging of meiosis II in *smg7-1* mutants. Time point 0 min corresponds to prometaphase II when chromosomes start condensing. Middle panel depicts PMCs in *smg7-1* PMCs arrested in aberrant anaphase II. Lower panel shows *smg7-1* PMCs that undergo brief telophase II before re-condensing again. Chromosomes were marked with HTA10:RFP. Images were taken every 5 min, scale bar 10 μm. Chromosomes within one PMC are indicated by dotted ovals.

The online version of this article includes the following video(s) for figure 7:

**Figure 7—video 1.** Time lapse imaging of chromosome segregation in meiosis I and meiosis II in a wild type plant in 2 min intervals.

https://elifesciences.org/articles/52546#fig7video1

**Figure 7—video 2.** Time lapse imaging of meiosis II and irregular anaphase II in *smg7-1* in 2 min intervals.

https://elifesciences.org/articles/52546#fig7video2

**Figure 7—video 3.** Time lapse imaging of meiosis II with brief telophase II and irregular anaphase III in *smg7-1* plant in 2 min intervals.

https://elifesciences.org/articles/52546#fig7video3

Nonetheless, this requires using higher power of excitation lasers to achieve a similar signal intensity, which may increase phototoxicity. In addition, a higher resolution will generate larger data volumes for downstream processing.

In this study, we demonstrate the power of LSFM to provide qualitatively novel information by capturing several cellular processes that were not previously studied by time-lapse microscopy. One of these is the restitution mitosis that leads to polyploidization of tapetum cells (*Oksala and Therman, 1977*). Previous cytogenetic analysis in Arabidopsis could not distinguish whether this polyploidization occurs through endomitosis or restitution mitosis (*Weiss, 2001*). We have also recorded the entire process of male meiosis in a single movie (*Figure 3*, *Figure 3—video 1*). The duration of male meiosis was recently determined to be 26 hr from late leptotene to telophase II (*Prusicki et al., 2019*). This study used landmark features of meiotic progression as anchors to compile information from multiple movies covering different segments of meiosis. Here we show that expression of ASY1 is an excellent marker for labeling meiotic cells. ASY1 starts being expressed about 5.5 hr ahead of late meiotic S-phase and is still detectable after tetrad formation. Based on the ASY1 signal, we estimate that the entire meiosis, from the end of S-phase until cytokinesis, lasts about 47 hr. This is

longer than 33 hr estimated from a DNA labeling experiment (*Armstrong et al., 2003*), which warrants further investigation whether the difference reflects used methodology, or whether in vitro cultivation during imaging delays meiotic progression. A unique feature of LSFM that distinguishes it from conventional confocal microscopy is its ability to simultaneously record cells in the entire organ. This opens new possibilities to study phenomena such as the synchrony of meiosis within and between anthers within a flower. Indeed, we recorded a time gradient of chromosome segregation within a loculus, as well as different timing of meiosis between short and long anthers. We have also performed live imaging of female meiosis. Plant male and female meiosis differ in many aspects including rates of recombination, and mutations in numerous meiotic genes have different phenotypic consequences in PMCs and MMCs, but the mechanisms underlying these differences are largely unknown (*Mercier et al., 2015*; *Sanders et al., 1999*; *Melamed-Bessudo et al., 2016*). Thus, our successful implementation of a live imaging protocol for female meiosis represents a major advancement for this neglected area of plant cell biology. In conclusion, we demonstrate that LSFM is well suited for live imaging of cellular processes occurring in Arabidopsis flowers and the method developed in this study will be broadly applicable in the research on plant sexual reproduction and flower development.

## Materials and methods

### Plant material and reporter constructs

*Arabidopsis thaliana* accession Columbia-0 harboring the reporter constructs described below were used in this study. The Arabidopsis wild-type planes expressing HTA10:RFP was kindly provided by Frederic Berger (*Dumur, 2019*). The construct was transferred to *smg7-1* mutants (*Riehs et al., 2008*) by crossing. Auxin response was measured using the *DR5::N7-Venus* reporter line (*Wabnik et al., 2013*). To generate ASY1:eYFP and H2B:mRuby2, the ASY1 (AT1G67370) and H2B (AT3G45980) genomic loci from *A. thaliana* accession Columbia as well as eYFP from pBlunt-EYFP-TAG (*Le Goff et al., 2020*) and mRuby2 CDS from pcDNA3-mRuby2 (*Lam et al., 2012*) (plasmid #40260; Addgene, www.addgene.com) were PCR amplified. Resulting amplicons were merged into one product, that is ASY1 together with eYFP and H2B together with mRuby2, in a subsequent PCR reaction, and inserted via *Sfi*I into the vector p35S-Nos-BM (dna-cloning-service.com). The resulting expression cassettes were sublconed via *Sfi*I into pLH7000 or pLH6000 (dna-cloning-service.com), respectively, and transformed into Arabidopsis *asy1* mutants (SALK_046272) by the floral dip method. To generate the PCNA:TagRFP reporter construct, the *AtPCNA1* gene (AT1G07370) was PCR amplified, cloned into pENTR/D-TOPO vector (Thermo Fisher Scientific), and then fused to TagRFP by transferring it into the binary vector pGWB659 by Gateway cloning system. The PCNA:TagRFP reporter was transformed into Arabidopsis by the floral dip method. Primers used in this study are indicated in the *Supplementary file 1A*.

### Sample preparation

Plants were grown in soil under long-day conditions (16 hr/8 hr light/dark regime at 21°C). Sample preparation for light sheet microscopy is illustrated in *Video 1*. Floral buds were detached from the main inflorescence bolt and their width was measured under a binocular microscope with a glass ruler (Dalekohledy a mikroskopy, www.dalekohledy.com). For imaging in open system, samples were prepared according to modified protocol (*Ovečka et al., 2015*). Sepals were carefully removed by tweezers and dissected buds were put into capillaries (glass capillary size 4, inner diameter 2.16 mm, Zeiss) containing medium (½ MS, 5% sucrose, pH 5.8) with 1% low melting point agarose (Sigma Aldrich). The capillary was fixed into the standard metal holder for the Zeiss Z1 microscope (Zeiss) and placed directly into the microscope chamber, which was filled with liquid medium (½ MS, 5% sucrose, pH 5.8). For imaging, the solidified medium with the floral bud was pushed out from the capillary in front of the objective. This open imaging system was used for short term imaging (up to 12 hr) and multiview 3D reconstruction. A closed cultivation system within FEP (Fluorinated Ethylene Propylene) tubes was used for long-term imaging. FEP tubes with an inner diameter of 2.80 mm, outer diameter 3.20 mm, and wall thickness of 0.20 mm (Wolf-Technik) were cut into ~4 cm pieces, boiled in the microwave, and sterilized in 70% ethanol. The tube was mounted onto the glass capillary. Medium and sample were placed inside the tube as described for the open system. The piston

was removed and the bottom of the FEP tube was sealed using a hot glue gun (Flying Tiger). The capillary with the attached FEP tube was then fixed in a metal holder and placed into the microscopic chamber which was filled with 6% glycerol (refractive index 1.33999, correcting the refractive index of 5% sucrose 1.3403). Glycerol was continually replaced using a peristaltic pump (GE Health-Care) with a flow rate of one chamber volume per hour. The temperature of the microscopy chamber was set up at 21°C. We routinely cultivated flowers in the closed system for up to five days with no exchange of the cultivation media.

## Imaging
Microscopy was performed with the Lightsheet Z1 (Zeiss) using detection objective 10x (0.5 NA W Plan-Apochromat 10x/0.5 M27 75 mm_4934000045), 2.5 zoom and Illumination objective 10x (Illumination Optics Lightsheet Z1 10x/0.2). We used two track imaging (frame fast setting) with 488 nm and 561 nm excitation lasers for GFP/YFP/autofluorescence and RFPs/mRuby2, respectively. The laser was blocked with the LBF 405/488/561/640 filter and the beam was split by an SBS LP 560 beam splitter for both tracks. The green track was recorded with the BP 505–545 and the red track with the BP 575–615 filters. To remove shadows, laser pivoting was always on. Light sheet thickness was set to the optimal value. Further details on imaging conditions and processing are provided in the *Supplementary file 1B* for each experiment.

## Image processing
*ZEN* software for Lightsheet Z1 (Zeiss) was used to subset data, create maximum intensity projections, add time stamps and scales, and export movies and figures. *ZEN* was also used to deconvolve selected data using Regularized inverse filter clip strength 2. *Fiji* was used to create maximum intensity projections (*Z project*), add scale bars and time stamps, and export selected figures and videos (*Schindelin et al., 2012*). Drift correction was done in *Fiji* with *Correct 3D Drift* (*Parslow et al., 2014*). Multi-view reconstruction and multi-view fusion were performed using the Multi-view Reconstruction plugin in Fiji (*Preibisch et al., 2010*). The labeled nuclei in one of the two imaged channels were used as fiduciaries to form the descriptors necessary to match and register the views. The parameters for this so-called 'segmentation-based registration' (*Schmied et al., 2014*) were as follows: Interest points (nuclei) were detected using Difference of Gaussian (sigma = 1.3075, threshold = 0.0244), registered by the Iterative-Closest Point (ICP) algorithm and aligned using regularized affine transformation model (lambda = 0.1). Following the registration, whose quality was accessed by examining the reconstructed volume in Fiji's BigDataViewer (*Pietzsch et al., 2015*), weighted-average image fusion with blending turned on was performed within the Multi-view Reconstruction plugin (*Preibisch et al., 2008*). Before the fusion, the reconstructed volume was cropped to a minimal size. Nevertheless, the size of the image data at full resolution still exceeded the typical RAM available on a Desktop computer. Therefore, the processing was outsourced to a computer node at the IT4Innovation supercomputing center in Ostrava, Czech Republic. In order to facilitate processing on a remote High Performance Computing (HPC) resource, Fiji plugins for remote cluster execution of SPIM multiview reconstruction were deployed (*Schmied et al., 2016*). After fusion, the data were saved as TIFF and ICS (Image Cytometry Standard) format and further processed and visualized using Imaris (Oxford Instruments). Automated PMC detection was done with the fused image in the .ims format in *Imaris* using spot detection wizard run on the green channel detecting ASY1. Videos were compressed by Handbrake (https://handbrake.fr/).

## Acknowledgements
We thank Frederic Berger for providing *HTA10:RFP* reporter line and Helene Robert-Boisivon for the *DR5::N7-Venus* line. This work was supported from European Regional Development Fund-Project 'REMAP' (No. CZ.02.1.01/0.0/0.0/15_003/0000479), the European Regional Development Fund in the IT4Innovations National Supercomputing Center—path to exascale project [No. CZ.02.1.01/0.0/0.0/16_013/0001791] within the Operational Programme Research, Development and Education, the German Federal Ministry of Education and Research (BMBF—FKZ 031B0188), and the IPK Gatersleben. We also acknowledge the core facility CELLIM of CEITEC supported by the Czech-BioImaging large RI project (LM2015062 funded by MEYS CR) for their support with obtaining scientific data

presented in this paper. Plant Sciences Core Facility of CEITEC MU is acknowledged for the cultivation of experimental plants used in this paper.

## Additional information

### Funding

| Funder | Grant reference number | Author |
| --- | --- | --- |
| European Regional Development Fund | REMAP CZ.02.1.01/0.0/0.0/15_003/0000479 | Karel Riha |
| European Regional Development Fund | CZ.02.1.01/0.0/0.0/16_013/0001791 | Pavel Tomancak |
| Bundesministerium für Bildung und Forschung | BMBF-FKZ 031B0188 | Stefan Heckmann |
| Leibniz Institute of Plant Genetics and Crop Plant Research | | Karel Riha |

The funders had no role in study design, data collection and interpretation, or the decision to submit the work for publication.

### Author contributions

Sona Valuchova, Conceptualization, Data curation, Formal analysis, Supervision, Funding acquisition, Investigation, Visualization, Methodology, Project administration; Pavlina Mikulkova, Conceptualization, Data curation, Formal analysis, Investigation, Visualization, Methodology; Jana Pecinkova, Resources, Formal analysis; Jana Klimova, Data curation, Formal analysis, Investigation, Methodology; Michal Krumnikl, Petr Bainar, Formal analysis; Stefan Heckmann, Resources, Funding acquisition; Pavel Tomancak, Resources, Formal analysis, Supervision, Funding acquisition, Methodology; Karel Riha, Conceptualization, Formal analysis, Supervision, Funding acquisition, Methodology, Project administration

### Author ORCIDs

Stefan Heckmann (iD) http://orcid.org/0000-0002-0189-8428
Pavel Tomancak (iD) http://orcid.org/0000-0002-2222-9370
Karel Riha (iD) https://orcid.org/0000-0002-6124-0118

### Decision letter and Author response

Decision letter https://doi.org/10.7554/eLife.52546.sa1
Author response https://doi.org/10.7554/eLife.52546.sa2

## Additional files

### Supplementary files

• Supplementary file 1. Supplementary material. (**A**) Oligonucleotides used in the study. (**B**) Overview of image processing.

• Transparent reporting form

### Data availability

Data was deposited to the Image Data Resource (https://idr.openmicroscopy.org) under accession number idr0077.

The following dataset was generated:

| Author(s) | Year | Dataset title | Dataset URL | Database and Identifier |
| --- | --- | --- | --- | --- |
| Sona Valuchova, Pavlina Mikulkova, Jana Pecinkova, | 2020 | Imaging plant germline differentiation within Arabidopsis flowers by light sheet microscopy | https://idr.openmicroscopy.org/search/?query=Name:77 | Image Data Resource, idr0077 |

Jana Klimova, Michal Krumnikl, Petr Bainar, Stefan Heckmann, Pavel Tomancak, Karel Riha

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
