## [Decision Letter]

**Acceptance summary:**

This paper describes a major technical advance for modern plant biology. Light sheet fluorescence microscopy (LSFM) is both much faster and gentler (low bleaching and phototoxicity) than point-scanning optical sectioning techniques such as confocal or multi-photon, and has emerged as a powerful tool for live imaging of cellular and developmental processes. This is the first study to successfully use LSFM to investigate developmental processes that occur beneath several cell layers in flowers, and that in itself is a major accomplishment. They visualized processes occurring in the buds and inside the floral tissues, including whole-organ growth, as well as mitotic and meiotic divisions in both the male and female germ lines. They could also monitor changes in protein expression and subcellular localization. This method is likely to be widely adopted to address numerous questions in plant reproductive biology.

**Decision letter after peer review:**

Thank you for submitting your article "Imaging plant germline differentiation within Arabidopsis flower by light sheet microscopy" for consideration by *eLife*. Your article has been reviewed by three peer reviewers, and the evaluation has been overseen by Sheila McCormick as the Reviewing Editor and Christian Hardtke as the Senior Editor. The following individuals involved in review of your submission have agreed to reveal their identity: Nathanaël Prunet (Reviewer #2); Peter Schloegelhofer (Reviewer #3); Anna A. Dobritsa (Reviewer #4).

The reviewers have discussed the reviews with one another and the Reviewing Editor has drafted this decision to help you prepare a revised submission.

Summary:

This paper describes a major technical advance in modern plant biology. Light sheet fluorescent microscopy is both much faster and gentler (low bleaching and phototoxicity) than point-scanning optical sectioning techniques like confocal or multi-photon, and has emerged as a powerful tool for live imaging of cellular and developmental processes. LSFM has been extensively used in animals. In plants it has been particularly successful in the study of roots. Early attempts (e.g. Ovecka et al., 2018) at using LSFM to study plant aerial tissues, and flowers met limited success, mostly due to the strong absorption of photosynthetic tissues, which limits the penetration of the light sheet. This is the first study to successfully use LSFM to investigate developmental processes that occur beneath several cell layers in flowers, and that in itself is a major accomplishment. They visualized a number of processes occurring in the buds and inside the floral tissues, including whole-organ growth. They were able to monitor mitotic and meiotic divisions in both the male and female germ lines and made some interesting new observations, such as the restitution mitosis in tapetum cells. They could also monitor changes in protein expression and subcellular localization. Recently a similar, but different, approach was presented in *eLife* (Prusicki et al., 2019) based on confocal microscopy. This other approach provides more resolution, but has two limitations: phototoxicity (short imaging windows) and limited field of view. The approach presented in this new study by the Riha lab is perfectly complementary to Prusicki et al. One might argue that the experimental set up is not really physiological, as the flower buds are removed from the inflorescence, sepals are removed (as well as a carpel valve when imaging female meiosis) and the specimen is embedded in low-density agarose. However, this is not different from most available protocols that have been used to do live cell imaging in flowers, and the authors made sure that, after dissection and embedding, the flower buds kept developing fairly normally.

Essential revisions:

1) Please provide further details for sample preparation and imaging. Because light sheet microscopy is less common than, for instance, confocal microscopy, and because it involves fairly challenging sample preparation, providing additional details would be helpful. For example, what capillaries were used (manufacturer, size, what were they made of)? How were the buds inserted into them and when was the solution added? Are the metal holder and the microscope chamber standard for the Zeiss Z1 microscope or were they chosen from a range of choices? Into what-size pieces were the FEP tubes cut? How were they mounted onto a glass capillary? In the "closed system", was the FEP tube with the sample sealed completely, from both ends? Perhaps adding illustrations or photos of the sample mounting process and of the imaging system to the supplementary materials would be helpful. How was the chamber kept at 21C – is it a feature of the system or some custom modification?

2) Please address magnification issues, and discuss why you chose a 10X objective. The images showing cellular details of meiosis (e.g. movements of chromosomes and formation of protein speckles) would greatly benefit from higher resolution. Some of the images in Figure 5 to 7 are hard to interpret (although videos definitely help). Would the use of higher-mag objectives – 60x, 100x – be compatible with the setup?

---

## [Author Response]

Essential revisions:1) Please provide further details for sample preparation and imaging. Because light sheet microscopy is less common than, for instance, confocal microscopy, and because it involves fairly challenging sample preparation, providing additional details would be helpful. For example, what capillaries were used (manufacturer, size, what were they made of)? How were the buds inserted into them and when was the solution added? Are the metal holder and the microscope chamber standard for the Zeiss Z1 microscope or were they chosen from a range of choices? Into what-size pieces were the FEP tubes cut? How were they mounted onto a glass capillary? In the "closed system", was the FEP tube with the sample sealed completely, from both ends? Perhaps adding illustrations or photos of the sample mounting process and of the imaging system to the supplementary materials would be helpful. How was the chamber kept at 21C – is it a feature of the system or some custom modification?

In order to provide more detailed description of the sample preparation and imaging, we have added more details in the Materials and methods section (subsection “Sample preparation”) and created new Video 1 that illustrates sample preparation and mounting for both open and closed imaging. Temperature maintenance is a feature of the Z1 system.

2) Please address magnification issues, and discuss why you chose a 10X objective. The images showing cellular details of meiosis (e.g. movements of chromosomes and formation of protein speckles) would greatly benefit from higher resolution. Some of the images in Figure 5 to 7 are hard to interpret (although videos definitely help). Would the use of higher-mag objectives – 60x, 100x – be compatible with the setup?

It is an important issue. The Zeiss Z1 light sheet microscope is usually sold with up to 40x objective and the manufacturer does not recommend purchasing higher magnification, although it is optionally available. The resolution in LSFM is not determined only by detection objectives, but also by the thickness and uniformity of the light sheet. It is technically challenging to produce thin and uniform light sheets in larger specimens (such as a flower), which limits use of high resolution objectives. Therefore, LSFM tends to be used on larger specimens where the desire for the highest possible resolutions is not required, but it is preferred to image fast (and to reduce phototoxicity) and still with reasonable quality.

Our primary goal was to develop a method for live cell imaging applications. Therefore, phototoxicity was a major issue for us. In our initial experiments we tested 10x, 20x and 40x lenses and found out that 10x lenses provide surprisingly good resolution (Author response image 1). While we can get slightly better resolution with 20x and 40x objectives, we have to use substantially higher power of excitation lasers and longer scans, which increases phototoxicity (eg with the 10x objective we used 5-10% of laser power, while 40-70% of laser power is required for the 40x lenses). Thus, use of higher resolution objectives may compromise temporal resolution as longer time windows between individual scans may be necessary to limit phototoxicty. Furthermore, scanning the same volume with higher resolution substantially increases amount of data, which is an important consideration for downstream image processing. We now include paragraph in Discussion to make these points (paragraph two).

**Author response image 1. respfig1:** Comparison of different objectives and zooming options. (**A**) Field of view when using objectives without zoom (0.36 zoom, upper panel) and with maximal zoom (2.5 zoom, lower panel). *ASY1:YFP HTA10:RFP* flowers were used in the experiment. (**B**) Comparison of resolution on 80x80um ROI between the objectives with the maximal zoom. ASY1:GFP merged with *HTA10:RFP* in upper panel, *HTA10:RFP* only is shown in lower panel.

The experiment in Author response image 1 is not best suited for illustrating fine differences in resolution between objectives, because at this stage ASY1 shows rather diffused localization. For the response to this review, we wanted to perform a rigorous comparison of different objectives on finer subcellular structures (condensed chromosomes, axial elements). Unfortunately, lasers on our LSFM microscope died just before the experiment and waiting for replacement would result in about two months delay in resubmitting the manuscript. Nevertheless, based on our experience we gained so far, we believe that use of 20x lenses may be beneficial in some applications where higher resolution is desired (e.g. studies focused on organization of axial elements or chromosome structures), but one has to consider increased phototoxicity as a trade off. We feel that there is no real point of using 40x lenses – this setting does not seem to provide benefits over conventional confocal microscopy, which still gives better resolution. Thus, choosing appropriate resolution depends on experimental questions and quality of reporter markers. The method we described here should not be perceived as a universal tool for solving all problems in plant cell biology. In many aspects, it is complementary tool to standard confocal microscopy (benefits of LSFM over standard confocal microscopy are discussed in the paper).

In regards to the interpretation of Figure 7: it is important to realize that the value of live imaging is in providing information in temporal context. Thus, the real informative datasets are supplementary videos. Figures in the main manuscript just attempt to illustrate key points obtained by analyzing the videos. It is true that resolution of chromosomes in Figure 7 is not great, but videos clearly show how chromosome condense and decondense over time. We have now indicated in the Figure 7 chromosomes that are located within one pollen mother cell, so readers can better track how compactness of chromosomes changes over time.